# ONE-PEACE: Exploring One General Representation Model Toward Unlimited Modalities

## Abstract

In this work, we propose ONE-PEACE, a highly extensible model with 4B parameters that seamlessly aligns and integrates representations across vision, audio, and language modalities. The ONE-PEACE architecture consists of shared self-attention layers, modality adapters and FFNs. This design allows for multi-modal fusion through self-attention layers, while also providing the flexibility to easily incorporate new modalities. Two modality-agnostic pretraining tasks, cross-modal aligning contrast and intra-modal denoising contrast, are developed to align the semantic space of different modalities and capture fine-grained details within each modality simultaneously. With the scaling-friendly architecture and tasks, ONE-PEACE has the potential to expand to unlimited modalities. Without utilizing any vision or language pretrained model for initialization, ONE-PEACE achieves new SOTAs across a wide range of uni-modal and cross-modal tasks. Furthermore, we show that ONE-PEACE possesses a strong emergent retrieval capability, enabling it to align modalities that are not paired in the training data.

## 1 Introduction

Representation models have received considerable attention in computer vision (Bao et al., 2021; He et al., 2022; 2020; Chen et al., 2020c; Caron et al., 2021; Oquab et al., 2023), speech processing (Schneider et al., 2019; Baevski et al., 2020; Chen et al., 2021; Hsu et al., 2021), natural language processing (Devlin et al., 2019; Liu et al., 2019; Clark et al., 2020; He et al., 2021), etc. Learning from large amounts of data, representation models demonstrate strong generalization ability in a wide range of downstream tasks. Furthermore, the explosive growth of large-scale language models (LLMs) has sparked an escalating appetite for representation models. Until recently, representation models have shown their bedrock role to unleash LLMs to understand, perceive, and interact with other modalities (e.g., vision) (OpenAI, 2023; Huang et al., 2023; Li et al., 2023b; Liu et al., 2023a).

Due to the distinct characteristics of different modalities, previous research mainly focuses on building uni-modal representation models with individual architectures and pretraining tasks. Despite achieving excellent results, these models face difficulties in effectively utilizing multi-modal data, which makes them challenging to extend to multi-modal tasks. With the development of unified architectures (Vaswani et al., 2017; Dosovitskiy et al., 2021; Jaegle et al., 2021) and efficient pretraining tasks (Devlin et al., 2019; Bao et al., 2021; He et al., 2022; Radford et al., 2021), recent works have achieved promising results in vision-language learning (Wang et al., 2022b; Alayrac et al., 2022; Yu et al., 2022; Wang et al., 2023; Li et al., 2022a; Chen et al., 2022) and audio-language learning (Guzhov et al., 2022; Wu et al., 2022; Elizalde et al., 2022). Nevertheless, there is still rare research on developing general models that can be applied to vision, audio, and language.

In this work, we propose ONE-PEACE, a vision-audio-language representation model with 4B parameters. The architecture of ONE-PEACE consists of multiple modality adapters and a modality fusion encoder. Each modality is equipped with an adapter for converting the raw inputs into features. The modality fusion encoder operates on the features with Transformer architecture. Each Transformer block contains a shared self-attention layer and multiple modality Feed Forward Networks (FFNs). The self-attention layer enables multimodal fusion through the attention mechanism, while the modality FFNs facilitate information extraction within modalities. With the clear division of labor in this architecture, extending new modalities only requires the injection of adapters and FFNs.

To pretrain ONE-PEACE, we design two modality-agnostic tasks, cross-modal contrastive learning and intra-modal denoising contrastive learning. Cross-modal contrastive learning includes both vision-language and audio-language contrastive learning to align the semantic spaces of vision, audio, and language. Intra-modal denoising contrastive learning computes the contrastive loss between the fine-grained masked features and visible features, e.g., image patches, language tokens, or audio waveform features. These tasks collaborate to enhance the model's finetuning performance while maintaining zero-shot retrieval capability.

Our contributions are summarized as follows:

- We propose ONE-PEACE, a highly extensible vision-audio-language representation model based on scaling-friendly architecture and modality-agnostic pretraining tasks. We highlight that ONE-PEACE has the potential to expand to unlimited modalities.

- Without using any vision or language pretrained model for initialization, ONE-PEACE achieves new SOTAs across a wide range of tasks, including semantic segmentation, audio-text retrieval, audio(-vision) classification, audio question answering, visual grounding, etc. To our knowledge, ONE-PEACE is the first representation model that achieves these results.

- ONE-PEACE exhibits strong emergent retrieval capabilities for aligning modalities that are not paired in the training data, thereby eliminating the need to collect paired data across different modalities.

## 2 RELATED WORK

Representation learning has been extensively studied in the fields of natural language processing, computer vision, speech processing, etc. Researchers develop various methods, including masked prediction (Devlin et al., 2019; Liu et al., 2019; He et al., 2022; Bao et al., 2021; Hsu et al., 2021), next unit prediction (Radford et al., 2018; Chen et al., 2020b; Ramesh et al., 2021; Borsos et al., 2022), contrastive learning (Gao et al., 2021; Chen et al., 2020c; He et al., 2020; Schneider et al., 2019; Baevski et al., 2020). These methods have been successfully used to train effective representation models that demonstrate strong generalization abilities in various downstream tasks.

Having witnessed the success of uni-modal representation learning, researchers have begun exploring building multi-modal representation models. One line of works uses contrastive learning to align different modalities (Radford et al., 2021; Jia et al., 2021; Yuan et al., 2021; Zhai et al., 2022; Li et al., 2022b). CLIP (Radford et al., 2021), the most representative model based on contrastive learning, has demonstrated strong transferability across diverse tasks (Zhou et al., 2022; Gao et al., 2023; Zhang et al., 2022a; Ramesh et al., 2022; Rombach et al., 2021). Other approaches jointly learn multimodal data with unified masked prediction tasks (Wang et al., 2023; Yang et al., 2022c; shi Zhu et al., 2022) but rely on external models (e.g., CLIP) to discretize the image or audio data. Data2vec (Baevski et al., 2022b;a) proposed a general self-supervised learning method that does not rely on external models. Although it successfully applied in vision, language, and audio modalities, it has not extended to multi-modal data. Recently, ImageBind (Girdhar et al., 2023) demonstrated impressive emergent retrieval capabilities across various modalities, but it trained separate models for each modality, which is challenging to employ in fine-grained multimodal tasks like visual grounding. Compared to previous works, the modality-agnostic tasks and unified architecture of ONE-PEACE make it not only easily adaptable to diverse modalities but also perform well in fine-grained uni-modal and cross-modal tasks.

## 3 METHOD

### 3.1 ARCHITECTURE

The model architecture of ONE-PEACE consists of three modality adapters and a modality fusion encoder. The overall architecture is shown in Figure 1.

**Modality Adapters.** Modality adapters are responsible for converting different raw signals into unified features. We designed three simple adapters for ONE-PEACE, namely: Vision Adapter, Audio Adapter, and Language Adapter. Vision Adapter utilizes hMLP (Touvron et al., 2022) to

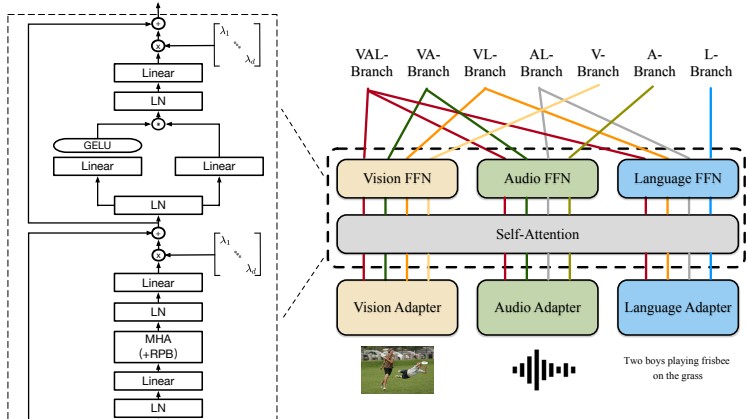

Figure 1: **The architecture of ONE-PEACE**. It can be disassembled into different branches to handle different tasks. For example, the vision adapter, self-attention layers, and vision FFNs can be combined into V-Branch to handle vision tasks.

convert images into image embeddings. The image representation is obtained by adding the image embeddings with absolute positional embeddings. Audio Adapter employs a convolutional feature extractor (Schneider et al., 2019) to transform audio into audio embeddings. Instead of absolute positional embeddings, a convolution layer is used to extract relative positional information (rahman Mohamed et al., 2019). Language Adapter uses byte-pair encoding (BPE) (Sennrich et al., 2016) to transform text into subword sequences, which are then embedded into text embeddings using the embedding layer. Absolute positional embeddings are used to supplement position information.

**Modality Fusion Encoder.**  Following previous works (Wang et al., 2022c;b; Yu et al., 2022; Wang et al., 2023), the modality fusion encoder is based on the Transformer architecture (Vaswani et al., 2017). We set up a self-attention layer and three modality feed-forward networks (FFNs) in each Transformer block. The shared self-attention layer enables the interaction between different modalities through the attention mechanism. The vision, audio, and language FFNs facilitate information extraction within modalities. We further make several improvements to stabilize training and improve performance. Sub-LayerNorm (Wang et al., 2022a) is incorporated into each Transformer block. Specifically, layer normalization is inserted before the input and output projections of each self-attention layer and FFN layer. The activation function in FFN is replaced with GeGLU (Shazeer, 2020) to improve model performance. For positional information, we introduce 1D relative position bias (Raffel et al., 2019) for text and audio, and 2D relative position bias for image (Dai et al., 2021). LayerScale (Touvron et al., 2021) is also used to adjust the output of each residual block. In our preliminary experiments, LayerScale is beneficial for stabilizing training and improving performance.

## 3.2 TASKS

The pretraining tasks of ONE-PEACE include cross-modal contrastive learning that endows the model with cross-modal retrieval capabilities, and intra-modal denoising contrastive learning that enables the model to achieve superior fine-tuning performances for downstream tasks. An illustration of the pretraining tasks is shown in Figure 2.

**Cross-Modal Contrastive Learning.**  Cross-modal contrastive learning effectively aligns the semantic spaces of different modalities. Given a sample pair $(S^1, S^2)$ of arbitrary modalities, we extract their features using the ONE-PEACE model. The outputs of the special tokens (e.g., vision class tokens) are regarded as global representations. Followed by a linear projection and normalization, we obtain the final representations $s^1$ and $s^2$. The loss function is shown below:

$$\mathcal{L}_{CL} = -\frac{1}{2N} \sum_{i=1}^{N} (\log \frac{\exp(s_i^1 s_i^2 / \sigma)}{\sum_{j=1}^{N} \exp(s_i^1 s_j^2 / \sigma)} + \log \frac{\exp(s_i^1 s_i^2 / \sigma)}{\sum_{j=1}^{N} \exp(s_j^1 s_i^2 / \sigma)}),$$

(1)

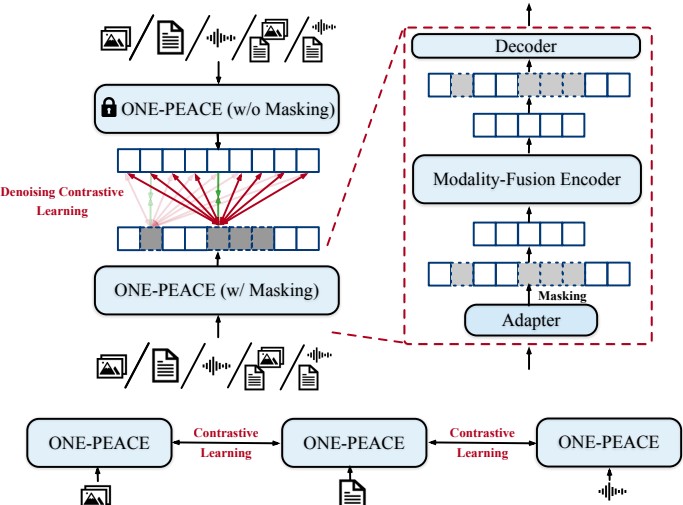

Figure 2: **The pretraining tasks of ONE-PEACE.** Intra-modal denoising contrastive learning encourages the features of the masked units (e.g., image patches or text tokens) close to the positive units (indicated by the green lines) and get away from the negative units (indicated by the red lines).

where $N$ is the batch size, $i, j$ are indexes within the batch, and $\sigma$ is a learnable temperature parameter (initialized to $0.07$). We apply cross-modal contrastive learning to image-text pairs and audio-text pairs, denoted by $\mathcal{L}_{CL-VL}$ and $\mathcal{L}_{CL-AL}$ respectively.

**Intra-Modal Denoising Contrastive Learning.** Cross-modal contrastive learning mainly focuses on aligning features between different modalities. However, it lacks emphasis on the learning of fine-grained details within modalities, leading to suboptimal performance in downstream tasks (Wei et al., 2022b). To address this issue, we further incorporate intra-modal denoising contrastive learning[1].

Given a sample of arbitrary modalities, we first encode it into an embedding sequence through the modality adapter. Then, we randomly mask some units (e.g., image patches or text tokens) and input the unmasked units to the modality fusion encoder (He et al., 2022). The encoded unmasked features are concatenated with the learnable mask tokens and fed to a lightweight Transformer decoder, which generates the masked features. We also use the ONE-PEACE model to encode the sample into target features without masking. Finally, we perform the contrastive loss between the masked features and target features, the loss function is shown below:

$$\mathcal{L}_{DCL} = -\frac{1}{N\hat{N}}\sum_{i=1}^{N}\sum_{j=1}^{\hat{N}}\log\frac{\exp(\hat{\boldsymbol{h}}_{ij}\cdot \mathrm{sg}(\boldsymbol{h}_{ij})/\tau)}{\sum_{m=1}^{N}\sum_{n=1}^{N}\exp(\hat{\boldsymbol{h}}_{ij}\cdot \mathrm{sg}(\boldsymbol{h}_{mn})/\tau)}, \qquad (2)$$

Where $\hat{\boldsymbol{h}}_{ij}$ and $\boldsymbol{h}_{ij}$ are the representations of the masked unit and target unit, respectively. $\hat{N}$ and $N$ are the numbers of the masked units and target units within a sample, respectively. $\mathrm{sg}(\cdot)$ is the stop gradient operation. $\tau$ is a constant temperature value, we set it to $0.4$. We apply this task to 5 types of data: image, audio, text, image-text pairs, and audio-text pairs. The loss are denoted by $\mathcal{L}_{DCL-V}$, $\mathcal{L}_{DCL-A}$, $\mathcal{L}_{DCL-L}$, $\mathcal{L}_{DCL-VL}$ and $\mathcal{L}_{DCL-AL}$, respectively. More details are provided in Appendix B.

### 3.3 TOWARD UNLIMITED MODALITIES

Based on the scaling-friendly architecture and modality-agnostic tasks, we designed an effective way to expand modalities step by step. Initially, the model is trained solely on image-text pairs. For each image-text pair, we calculate $\mathcal{L}_{CL-VL}$ and $\mathcal{L}_{DCL-VL}$, as well as $\mathcal{L}_{DCL-V}$ and $\mathcal{L}_{DCL-L}$ using the image and text separately. Subsequently, the model is trained on audio-text pairs, with only the audio-related parameters being updated. The remaining parameters, including self-attention layers,

---

[1]Intra-modal denoising contrastive learning is similar to (Yi et al., 2022), but extends to more modalities.

Table 1: **System-level comparisons of image classification on ImageNet-1k.**

| Method | Enc. #Params | Patch Size | Image Size | acc |
|---|---|---|---|---|
| FD-SwinV2-G | 3.0B | $16 \times 16$ | $336^2$ | 89.4 |
| InternImage | 1.08B | $16 \times 16$ | $640^2$ | 89.2 |
| BEiT-3 | 1.01B | $14 \times 14$ | $336^2$ | 89.6 |
| EVA | 1.01B | $14 \times 14$ | $560^2$ | 89.7 |
| **ONE-PEACE** | 1.52B | $16 \times 16$ | $384^2$ | 89.6 |
| **ONE-PEACE** | 1.52B | $16 \times 16$ | $512^2$ | **89.8** |
| *methods using extra privately collected data* | | | | |
| RevCol-H | 2.16B | - | $640^2$ | 90.0 |
| ViT-G | 1.84B | $14 \times 14$ | $518^2$ | 90.5 |
| CoCa | 1.01B | $18 \times 18$ | $576^2$ | 91.0 |

Table 2: **System-level comparisons of semantic segmentation on ADE20k.** mIoU$^{ss}$ means single-scale inference result while mIoU$^{ms}$ means multi-scale.

| Method | Enc. #Params | Crop Size | mIoU$^{ss}$ | mIoU$^{ms}$ |
|---|---|---|---|---|
| RevCol-H | 2.16B | $640^2$ | 60.4 | 61.0 |
| FD-SwinV2-G | 3.00B | $896^2$ | - | 61.4 |
| ViT-Adapter | 571M | $896^2$ | 61.2 | 61.5 |
| EVA | 1.01B | $896^2$ | 61.5 | 62.3 |
| BEiT-3 | 1.01B | $896^2$ | 62.0 | 62.8 |
| InternImage | 1.08B | $896^2$ | **62.5** | 62.9 |
| **ONE-PEACE** | 1.52B | $896^2$ | 62.0 | **63.0** |

Table 3: **System-level comparisons of object detection and instance segmentation on MSCOCO.** The reported results are obtained by directly fine-tuning the models on MSCOCO, without intermediate fine-tuning on Objects365.

| Method | Detector | #Params | AP$^{box}$ | AP$^{mask}$ |
|---|---|---|---|---|
| ViT-Adapter | HTC++ | 401M | 58.8 | 51.1 |
| ViTDet | Cascade | 692M | 60.4 | 52.0 |
| RevCol-H | HTC++ | 2.41B | **61.1** | **53.0** |
| **ONE-PEACE** | Cascade | 1.59B | 60.4 | 52.9 |

Table 4: **System-level comparisons of video action recognition on Kinetics-400.**

| Method | Backbone | Input Size | Top-1 | Top-5 |
|---|---|---|---|---|
| VATT | ViT-L | $32 \times 320^2$ | 82.1 | 95.5 |
| Florance | Co-Swin-H | N/A $\times 384^2$ | 86.5 | 97.3 |
| SwinV2 | Swin-G | N/A $\times 384^2$ | 86.8 | - |
| VideoMAE | ViT-H | $32 \times 320^2$ | 87.4 | 97.6 |
| VideoMAE V2 | ViT-H | $32 \times 320^2$ | 87.4 | 97.6 |
| CoCa (frozen) | ViT-g | $16 \times 576^2$ | 88.0 | - |
| ViT-22B (frozen) | ViT-22B | $128 \times 224^2$ | 88.0 | - |
| **ONE-PEACE** (frozen) | ViT-g | $16 \times 256^2$ | 88.0 | **97.8** |
| **ONE-PEACE** (frozen) | ViT-g | $32 \times 256^2$ | **88.1** | **97.8** |

are frozen. Although the model is not trained on image-audio pairs, the semantic space between vision and audio is still aligned by using language as the anchor. The motivation behind this training process is that we can align new modalities based on existing modalities, while not compromising the performance of the existing ones. As a result, ONE-PEACE can expand to unlimited modalities.

# 4 PRETRAINING DETAILS

**Pretraining Datasets.** The pretraining datasets of ONE-PEACE are divided into two parts: image-text pairs and audio-text pairs. For image-text pairs, we use LAION-2B (Schuhmann et al., 2022), a dataset obtained by web crawling. For audio-text pairs, we collect a large amount of open-source environmental sound datasets. To ensure reproducibility, all pretraining datasets are publicly available. We provide more details about the pretraining datasets in Appendix C.1.

**Pretraining Settings.** ONE-PEACE is a giant-size model with 4B parameters. The model weights of ONE-PEACE are randomly initialized at the beginning, except for the audio feature extractor of Audio adapter, for which we use the weights of WavLM's feature extractor (Chen et al., 2021) for initialization. We find that incorporating this feature extractor significantly improves the model performance. More details are provided in Appendix C.2 and E.1.

# 5 EXPERIMENTS

## 5.1 RESULTS ON VISION TASKS

We transfer ONE-PEACE to various mainstream vision benchmarks, including image classification, semantic segmentation, object detection, instance segmentation, and video action recognition. We provide the implementation details in Appendix D.1.

**Image Classification.** In our experiments, we assess the image classification transfer performance of ONE-PEACE using the ImageNet-1K (Russakovsky et al., 2015) dataset. As demonstrated in Table 1, ONE-PEACE obtains **89.8** top-1 accuracy on ImageNet with less token length $(image\_size/patch\_size)^2$. Note that FD-SwinV2-G (Liu et al., 2022b), BEiT-3 (Wang et al., 2023), and EVA (Fang et al., 2023) all rely on the assistance of an external CLIP model for pretraining, while

Table 5: **Experimental results on audio-text retrieval.** ONE-PEACE significantly outperforms baselines by a large margin.

| Method | AudioCaps | | | | | | Clotho | | | | | |
| | Text → Audio | | | Audio → Text | | | Text → Audio | | | Audio → Text | | |
| | R@1 | R@5 | R@10 | R@1 | R@5 | R@10 | R@1 | R@5 | R@10 | R@1 | R@5 | R@10 |
|---|---|---|---|---|---|---|---|---|---|---|---|---|
| MMT (Koepke et al., 2021) | 36.1 | 72.0 | 84.5 | 39.6 | 76.8 | 86.7 | 6.7 | 21.6 | 33.2 | 7.0 | 22.7 | 34.6 |
| ML-ACT (Mei et al., 2022) | 33.9 | 69.7 | 82.6 | 39.4 | 72.0 | 83.9 | 14.4 | 36.6 | 49.9 | 16.2 | 37.6 | 50.2 |
| CLAP-HTSAT (Deshmukh et al., 2022) | 34.6 | 70.2 | 82.0 | 41.9 | 73.1 | 84.6 | 16.7 | 41.1 | 54.1 | 20.0 | 44.9 | 58.7 |
| TAP (Xin et al., 2023) | 36.1 | 72.0 | 85.2 | 41.3 | 75.5 | 86.1 | 16.2 | 39.2 | 50.8 | 17.6 | 39.6 | 51.4 |
| LAION-CLAP (Wu et al., 2023) | 35.1 | 71.5 | 83.6 | 45.8 | 80.9 | 91.6 | 18.2 | 42.5 | 54.4 | 25.7 | 51.5 | 63.4 |
| **ONE-PEACE** | **42.5** | **77.5** | **88.4** | **51.0** | **81.9** | **92.0** | **22.4** | **49.0** | **62.7** | **27.1** | **52.3** | **65.4** |

Table 6: **Experimental results on audio classification and audio question answering (AQA).** "ZS" is short for zero-shot results, "FT" is short for fine-tuning results. *We use the official code of LAION-CLAP to reproduce the result on VGGSound.

| Method | ESC-50 ZS | FSD50K FT | VGGSound FT | AQA / AVQA FT |
|---|---|---|---|---|
| Previous SOTA | 91.0 (Wu et al., 2023) | 65.6 (Koutini et al., 2021) | 67.1 (Huang et al., 2022) | 83.5 (Yang et al., 2022a) / 90.2 (Li et al., 2023a) |
| Wav2CLIP (Jia et al., 2021) | 41.4 | 43.1 | 46.6 | - |
| ImageBind (Girdhar et al., 2023) | 66.9 | - | - | - |
| AudioCLIP (Yao et al., 2022) | 69.4 | - | - | - |
| CLAP (Elizalde et al., 2022) | 82.6 | 58.6 | - | - |
| LAION-CLAP (Wu et al., 2023) | 91.0 | 46.2 | 55.1* | - |
| **ONE-PEACE** | **91.8** | **69.7** | **68.3** | **86.2 / 92.2** |

ONE-PEACE is trained from scratch without the help of external models. Even so, ONE-PEACE is able to achieve better results, which demonstrates its strong transferability.

**Semantic Segmentation.** We experiment on ADE20k (Zhou et al., 2016) using ViT-Adapter (Chen et al., 2023) for task adaptation and Mask2Former (Cheng et al., 2022) as the segmentation head. Following common practice, we first fine-tune the segmentation head on COCO-stuff (Caesar et al., 2018) then fine-tune on ADE20k. As demonstrated in Table 2, ONE-PEACE establishes a new state-of-the-art, achieving a mean Intersection over Union (mIoU) of **63.0**. This result indicates that ONE-PEACE exhibits exceptional transferring performance in the domain of dense prediction tasks.

**Object Detection and Instance Segmentation.** We perform fine-tuning experiments on the COCO 2017 (Lin et al., 2014) dataset. We employ the ONE-PEACE backbone and use the ViTDet (Li et al., 2022c) with Cascade Mask-RCNN architecture. Soft-NMS (Bodla et al., 2017) is used during the inference stage. As illustrated in Table 3, the instance-level transfer capabilities of ONE-PEACE exhibit a performance that is on par with the current state-of-the-art methods.

**Video Action Recognition.** We benchmark ONE-PEACE on Kinetics 400 (Carreira & Zisserman, 2017) dataset for video action recognition. Following AIM (Yang et al., 2023), we keep the model frozen and add several MLP adapters in each transformer layer. We use I3D (Carreira & Zisserman, 2017) head as the classification layer. As demonstrated in Table 4, without fine-tuning the full encoder, ONE-PEACE could achieve **88.1** top-1 accuracy, even outperforming CoCa (Yu et al., 2022) which is pre-trained on private data, and ViT-22B (Dehghani et al., 2023) with 14x more parameters.

## 5.2 RESULTS ON AUDIO-RELATED TASKS

We evaluate ONE-PEACE on various audio-related tasks, including audio-text retrieval, audio(-vision) classification, and audio question answering. Implementation details are provided in Appendix D.2.

**Audio-Text Retrieval.** For audio-text retrieval, we conduct experiments on both AudioCaps (Kim et al., 2019) and Clotho (Drossos et al., 2019) datasets. As shown in Table 5, ONE-PEACE achieves SOTA results on these datasets, outperforming the previous audio representation model by a large margin. On AudioCaps, ONE-PEACE achieves 21.1% improvement on R@1 in text-to-audio retrieval and 11.4% improvement on R@1 in audio-to-text retrieval. On Clotho, ONE-PEACE achieves 23.1% improvement on R@1 in text-to-audio retrieval and 5.4% on R@1 in audio-to-text retrieval.

Table 7: **Experimental results on image-text retrieval.** For a fair comparison, the reported results of BEiT-3 are obtained by directly fine-tuning on downstream tasks without intermediate fine-tuning.

| Method | COCO (5K test set) | | | | | | Flickr30K (1K test set) | | | | | |
| | Image → Text | | | Text → Image | | | Image → Text | | | Text → Image | | |
| | R@1 | R@5 | R@10 | R@1 | R@5 | R@10 | R@1 | R@5 | R@10 | R@1 | R@5 | R@10 |
|---|---|---|---|---|---|---|---|---|---|---|---|---|
| *Zero-shot Setting* | | | | | | | | | | | | |
| CLIP (Radford et al., 2021) | 58.4 | 81.5 | 88.1 | 37.8 | 62.4 | 72.2 | 88.0 | 98.7 | 99.4 | 68.7 | 90.6 | 95.2 |
| ALIGN (Jia et al., 2021) | 58.6 | 83.0 | 89.7 | 45.6 | 69.8 | 78.6 | 88.6 | 98.7 | 99.7 | 75.7 | 93.8 | 96.8 |
| FILIP (Yao et al., 2022) | 61.3 | 84.3 | 90.4 | 45.9 | 70.6 | 79.3 | 89.8 | 99.2 | 99.8 | 75.0 | 93.4 | 96.3 |
| Florence (Yuan et al., 2021) | 64.7 | 85.9 | - | 47.2 | 71.4 | - | 90.9 | 99.1 | - | 76.7 | 93.6 | - |
| CoCa (Yu et al., 2022) | **66.3** | **86.2** | 91.8 | **51.2** | **74.2** | **82.0** | **92.5** | **99.5** | **99.9** | **80.4** | **95.7** | **97.7** |
| **ONE-PEACE** | 64.7 | 86.0 | **91.9** | 48.0 | 71.5 | 79.6 | 90.9 | 98.8 | 99.8 | 77.2 | 93.5 | 96.2 |
| *Fine-tuning Setting* | | | | | | | | | | | | |
| ALIGN (Jia et al., 2021) | 77.0 | 93.5 | 96.9 | 59.9 | 83.3 | 89.8 | 95.3 | 99.8 | 100.0 | 84.9 | 97.4 | 98.6 |
| FILIP (Yao et al., 2022) | 78.9 | 94.4 | 97.4 | 61.2 | 84.3 | 90.6 | 96.6 | 100.0 | 100.0 | 87.1 | 97.7 | 99.1 |
| Florence (Yuan et al., 2021) | 81.8 | 95.2 | - | 63.2 | 85.7 | - | 97.2 | 99.9 | - | 87.9 | 98.1 | - |
| BEiT-3 (Wang et al., 2023) | 82.7 | 96.0 | 98.2 | 65.1 | **86.6** | **92.3** | 97.5 | 99.9 | **100.0** | 89.1 | **98.6** | **99.3** |
| **ONE-PEACE** | **84.1** | **96.3** | **98.3** | **65.4** | 86.3 | 91.9 | **97.6** | **100.0** | **100.0** | **89.6** | 98.0 | 99.1 |

Table 8: **Experimental results on** 3 **visual grounding datasets**: RefCOCO, RefCOCO+, RefCOCOg. ONE-PEACE achieves state-of-the-are results without using additional visual grounding datasets.

| Method | RefCOCO | | | RefCOCO+ | | | RefCOCOg | |
| | val | testA | testB | val | testA | testB | val-u | test-u |
|---|---|---|---|---|---|---|---|---|
| MDETR (Kamath et al., 2021) | 86.75 | 89.58 | 81.41 | 79.52 | 84.09 | 70.62 | 81.64 | 80.89 |
| UNICORN (Yang et al., 2022b) | 88.29 | 90.42 | 83.06 | 80.30 | 85.05 | 71.88 | 83.44 | 83.93 |
| X-VLM (Zeng et al., 2021) | - | - | - | 84.51 | 89.00 | 76.91 | - | - |
| Grounding-DINO (Liu et al., 2023b) | 90.56 | 93.19 | 88.24 | 82.75 | 88.95 | 75.92 | 86.13 | 87.02 |
| FIBER (Dou et al., 2022) | 90.68 | 92.59 | 87.26 | 85.74 | 90.13 | 79.38 | 87.11 | 87.32 |
| OFA (Wang et al., 2022b) | 92.04 | 94.03 | 88.44 | 87.86 | 91.70 | 80.71 | 88.07 | 88.78 |
| **ONE-PEACE** | **92.58** | **94.18** | **89.26** | **88.77** | **92.21** | **83.23** | **89.22** | **89.27** |

**Audio Classification & Audio(-Video) Question Answering.** As shown in Table 6, ONE-PEACE achieves SOTA results on a series of audio classification and audio(-video) question answering datasets. Notably, despite not pretraining on vision-audio pair data, ONE-PEACE still achieves a high score of **68.3** on the video-audio classification dataset VGGsound, surpassing the previous SOTA by 1.2. These results demonstrate the superior ability of ONE-PEACE on audio-related tasks.

## 5.3 RESULTS ON VISION-LANGUAGE TASKS

We conduct experiments on image-text retrieval, visual grounding, and vision-language understanding[2]. The implementation details are provided in Appendix D.3.

**Image-Text Retrieval.** Table 7 presents the performance of ONE-PEACE and baseline models on the image-text retrieval task. Under the fine-tuning setting, ONE-PEACE achieves the best performance in both MSCOCO Lin et al. (2014) and Flickr30K Young et al. (2014) test sets. This indicates that combining both cross-modal and intra-modal contrastive loss can lead to better performance in downstream retrieval tasks. Under the zero-shot setting, ONE-PEACE can achieve better or competitive performance compared to previous dual-encoder models like CLIP and Florence. Notice that the results of ONE-PEACE are inferior to CoCa, which might be because ONE-PEACE only trained on 6.4 billion image-text pairs while CoCa trained on up to 32 billion image-text pairs.

**Visual Grounding.** To evaluate the capability of visual grounding, we conduct experiments on RefCOCO/+/g datasets (Yu et al., 2016; Mao et al., 2016). Table 8 presents the results of ONE-PEACE and baseline models. It is worth noting that previous SOTA OFA use additional visual grounding datasets for training (i.e., Visual Genome (Krishna et al., 2017)). Without introducing additional labeled data, ONE-PEACE still achieves new SOTA results on the 3 datasets. In addition, ONE-PEACE can perform accurate visual grounding on the out-of-domain images, and we provide quality examples in Appendix F.

---

[2]We provide the results on vision-language understanding tasks in Appendix E.2

Table 9: **Ablation experiments on model structures.** "Share ATTN & FFN" means share both self-attention layers and FFN layers. "No Share" means separate both self-attention layers and FFN layers. "Share FFN" means separate self-attention layers and share FFN layers. "Share ATTN" means share self-attention layers and separate FFN layers, which is the default setting of ONE-PEACE.

| Structure | COCO zero-shot (5k test set) | | | | | | IN-1K |
|---|---|---|---|---|---|---|---|
| | Image → Text | | | Text → Image | | | |
| | R@1 | R@5 | R@10 | R@1 | R@5 | R@10 | ZS |
| Share ATTN & FFN | 33.96 | 60.92 | 72.24 | 22.51 | 46.13 | 57.63 | 42.21 |
| No Share | 29.08 | 54.30 | 65.56 | 18.98 | 40.34 | 51.44 | 37.73 |
| Share FFN | 27.36 | 52.66 | 64.36 | 18.08 | 38.68 | 49.84 | 33.98 |
| Share ATTN | **35.94** | **62.52** | **72.78** | **23.87** | **47.80** | **59.38** | **43.24** |

Table 10: **Ablation studies of intra-modal denoising contrastive learning.** "CL" is cross-modal contrastive learning, "DCL-L", "DCL-V", and "DCL-VL" means applying intra-modal denoising contrastive learning to language, vision, and vision-language data, respectively.

| CL | DCL-L | DCL-V | DCL-VL | COCO zero-shot (5k test set) | | | | | | IN-1K | |
|---|---|---|---|---|---|---|---|---|---|---|---|
| | | | | Image → Text | | | Text → Image | | | | |
| | | | | R@1 | R@5 | R@10 | R@1 | R@5 | R@10 | ZS | FT |
| ✓ | | | | 35.94 | 62.52 | 72.78 | 23.87 | 47.80 | 59.38 | 43.24 | 82.20 |
| ✓ | ✓ | | | 37.02 | 63.78 | 73.96 | 23.63 | 47.87 | 59.22 | 43.69 | 81.99 |
| ✓ | | ✓ | | 38.88 | 65.34 | 75.76 | 26.05 | 49.78 | 61.26 | 45.94 | 83.32 |
| ✓ | ✓ | ✓ | | 39.00 | 65.64 | 76.30 | 25.85 | 50.06 | 61.80 | 45.54 | 83.33 |
| ✓ | ✓ | ✓ | ✓ | **39.94** | **65.94** | **76.72** | **26.94** | **51.38** | **62.81** | **46.41** | **83.75** |

## 5.4 ABLATION STUDY

For the following ablation experiments, we utilize VIT-B/16 as the model backbone. The model is trained for 20 epochs with a batch size of 4096. We randomly selected 20 million image-text pairs from Laion-2B as the pretraining dataset.

**Ablation on Model Structures.** We first investigate the effects of sharing or separating different modules. As shown in Table 9, sharing all layers performs better than not sharing, indicating that utilizing a single Transformer can effectively align vision and language modalities. Furthermore, it is more beneficial to separate the FFN layer instead of sharing it. This implies that separating the FFN layer enhances the model's ability to extract modality-specific information, leading to more accurate representations. We also find separating the self-attention layer and sharing the FFN layer resulted in the poorest performance, suggesting that the self-attention layer plays a more significant role in modality alignment than the FFN layer. In addition, "Share ATTN" exhibits faster convergence speed compared to other architectures, we demonstrate the training curves in Appendix E.3.

**Effects of Intra-modal Denoising Contrastive Learning.** We examine the effects of intra-modal denoising contrastive learning (DCL). As shown in Table 10, applying DCL to language data (DCL-L) improves the model's performance in text retrieval. Furthermore, applying DCL to vision data (DCL-V) enhances the model's cross-modal retrieval ability, as well as fine-tuning performance in image classification. By applying DCL to vision-language data (DCL-VL), ONE-PEACE achieves the best results in all the metrics. These results demonstrate that DCL complements cross-modal contrastive learning to achieve outstanding performance in the downstream fine-tuning, while also enhancing the model's capability for zero-shot cross-modal retrieval. Noting that we can also combine DCL with FLIP (Li et al., 2022b) to improve training efficiency, we leave this to future research.

**Ablation on Different Denoising Losses.** We conduct a systematic comparison of different denoising losses, including the smooth L1 loss used in Baevski et al. (2022b); Dong et al. (2022), the L2 loss used in Baevski et al. (2022a); Liu et al. (2022a), the cosine loss used in Wei et al. (2022a); Fang et al. (2023); Zhang et al. (2022b), and the denoising contrastive loss used in this paper. As shown in Table 11, different types of denoising loss can improve the model's performance in both cross-modal retrieval and image classification tasks. Among them, denoising contrastive loss has the greatest improvement in terms of all the metrics, which indicates that denoising contrastive loss is more compatible with cross-modal contrastive loss than other denoising losses.

Table 11: **Ablation studies of different denoising losses.** Among all the masked losses, denoising contrastive loss shows the greatest performance improvement in all the metrics.

| Denoising Loss | COCO zero-shot (5k test set) | | | | | | IN-1K | |
| | Image → Text | | | Text → Image | | | | |
| | R@1 | R@5 | R@10 | R@1 | R@5 | R@10 | ZS | FT |
|---|---|---|---|---|---|---|---|---|
| None | 35.94 | 62.52 | 72.78 | 23.87 | 47.80 | 59.38 | 43.24 | 82.20 |
| Smooth L1 Loss | 36.72 | 63.86 | 74.30 | 24.37 | 48.17 | 59.86 | 44.36 | 82.50 |
| L2 Loss | 37.46 | 64.80 | 74.46 | 25.81 | 49.64 | 61.80 | 45.78 | 83.15 |
| Cosine Loss | 38.28 | 65.80 | 75.18 | 25.47 | 50.02 | 61.59 | 45.15 | 83.07 |
| Denoising Contrastive Loss | **39.94** | **65.94** | **76.72** | **26.94** | **51.38** | **62.81** | **46.41** | **83.75** |

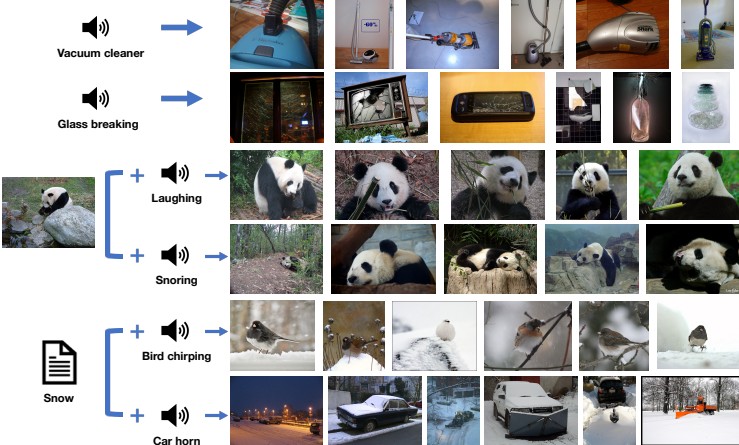

Figure 3: Examples of emergent zero-shot retrieval. ONE-PEACE is capable of aligning modalities and modality combinations, where there are no paired data of the modalities in the pretraining dataset. The images are retrieved from ImageNet-1K and MSCOCO.

## 5.5 EMERGENT ZERO-SHOT RETRIEVAL

In our pretraining, we exclusively align other modalities with text which plays as an intermediary role. We assume that our model is able to align those modalities that are not paired in the training data. For example, ONE-PEACE should be able to align image and audio. Thus, we conduct experiments on the retrieval of those modalities to assess the emergent zero-shot capabilities (Girdhar et al., 2023).

To be more specific, we evaluate the audio-to-image, audio+image-to-image, and audio+text-to-image retrieval abilities and demonstrate case studies in Figure 3. The first two cases demonstrate the emergent capability of uni-modal retrieval, while the other cases show that of the retrieval of image based on multimodal inputs. Specifically, we find that ONE-PEACE is able to retrieve images that contain elements concerning inputs of different modalities, e.g., the model uses the text "snow" and the sound of bird chirping to retrieve the images of birds in the snow. These examples demonstrate that ONE-PEACE has strong potential in emergent zero-shot capabilities. This indicates that for a universal representation model, there is no need to learn all pairing relationships between modalities, but instead it is sufficient for modalities to be aligned to an intermediary one. We provide more quality examples in Appendix G.

## 6 CONCLUSION

In this work, we explore a scalable way for building a general representation model across different modalities. Based on the flexible architecture and modality-agnostic pretraining tasks, we release ONE-PEACE, a general representation model that seamlessly aligns and integrates representations across vision, audio, and language modalities. The experimental results demonstrate that ONE-PEACE achieves SOTA results in a wide range of tasks, including semantic segmentation, audio-text retrieval, audio(-vision) classification, audio question answering, image-text retrieval, visual grounding, etc. Furthermore, we show that ONE-PEACE possesses a strong emergent zero-shot retrieval capability, enabling it to align modalities that are not paired in the training data.

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

## A    DISCUSSION

Recently, there has been a surge in the development of multimodal representation models that aim to integrate multiple modalities into a unified semantic space. (Girdhar et al., 2023; Xue et al., 2023; Liang et al., 2022). Notably, HighMMT (Liang et al., 2022) has pushed the boundaries by extending a single model to encompass 10 modalities. Compared to HighMMT, we take a more gentle design to gradually extend ONE-PEACE to multiple modalities. This design allows us to add new modalities (audio) into ONE-PEACE without affecting its performance on existing modalities (vision and language), and even enables the achievement of SOTA results in tasks related to the newly added modalities. Despite these successes, we have not demonstrated the capability of ONE-PEACE in other modalities, especially heterogeneous ones, which needs to be addressed in future work. Meanwhile, another emerging trend involves the fusion of multimodal representation models with large-scale language models (LLMs) (Li et al., 2023b; Liu et al., 2023a; Su et al., 2023; Zhang et al., 2023; Zhao et al., 2023; Han et al., 2023). This integration aims to equip LLMs with the ability to comprehend other modalities. It is crucial to recognize that as a general representation model, ONE-PEACE can also be integrated into LLMs to provide multimodal capabilities. Unlike representation models like CLIP (Radford et al., 2021), CLAP (Elizalde et al., 2022), and ImageBind (Girdhar et al., 2023), ONE-PEACE allows the multimodal features to interact before entering LLMs, and we expect this early fusion approach will yield more precise multimodal features for LLMs. In the future, we will further improve ONE-PEACE from the following aspects:

- Introducing more modalities, especially heterogeneous modalities like IMU, to further demonstrate that ONE-PEACE can extend to infinite modalities.
- By combining LLMs, creating a more powerful multimodal language model.

## B    APPLY DCL TO DIFFERENT TYPES OF DATA

We apply intra-modal denoising contrastive learning (DCL) to 5 types of data: image, audio, text, image-text pairs, and audio-text pairs. For image, we randomly mask $75\%$ patches, the loss function used for this type of data is denoted by $\mathcal{L}_{DCL-V}$. For audio, we sample $p = 0.11$ of all time-steps to be starting indices and mask the subsequent 5 time-steps. We manually adjust the mask rate to make approximately $55\%$ of all time-steps to be masked for each sample. The loss function is denoted by $\mathcal{L}_{DCL-A}$. For text, we randomly mask $15\%$ tokens of the text sequence, the loss function is denoted by $\mathcal{L}_{DCL-L}$. For image-text pairs, we randomly mask $68.75\%$ patches of the image and $40\%$ tokens of the text. The unmasked patches and tokens are concatenated together and encoded as masked features. The original image patches and text tokens are also concatenated together and encoded as target features. We then perform contrastive loss on the image patches and text tokens respectively, the average of these two losses is denoted by $\mathcal{L}_{DCL-VL}$. For audio-text pairs, we randomly mask $45\%$ time-steps of the audio waveform and $40\%$ tokens of the text. The loss is similar to the above one, we denote it by $\mathcal{L}_{DCL-AL}$.

## C    PRETRAINING DETAILS

### C.1    PRETRAINING DATASETS

For image-text pairs, we use LAION-2B (Schuhmann et al., 2022), a dataset obtained by web crawling that may contain some noisy pairs. To improve the data quality, we apply several pre-processing steps, including removing images with an aspect ratio greater than 3.5, removing images with the shortest side less than 128, and removing images with a CLIP score less than 0.3. We also remove texts containing non-English or emoji characters, as well as texts with lengths less than 3 or greater than 512. After these steps, we retain about 1.5 billion image-text pairs.

For audio-text pairs, we mainly use the environmental sound datasets processed by (Wu et al., 2023). Specifically, for some datasets that only contain tags, (Wu et al., 2023) uses a pretrained language model T5 (Raffel et al., 2019) to rewrite these tags into captions. We also perform simple cleaning on the data, which involves removing samples with text lengths less than 3 or greater than 512, as well as texts containing non-English or emoji characters. Ultimately, we obtain about 2.4 million

audio-text pairs, with a total duration of around 8,000 hours. Table 12 presents the environmental sound datasets utilized by ONE-PEACE.

Table 12: **Statistics on the environmental sound datasets.** All datasets are publicly available.

| Dataset | Number of Samples | Duration | T5 Augmentation |
|---|---|---|---|
| Epidemic Sound | 75618 | 220.40 hrs | Yes |
| AudioCaps (Kim et al., 2019) | 49494 | 135.56 hrs | No |
| AudioSet (Gemmeke et al., 2017) | 1910918 | 5263.23 hrs | Yes |
| AudioStock | 9552 | 40.31 hrs | No |
| Clotho (Drossos et al., 2019) | 3839 | 23.99 hrs | No |
| FreeSound (Font et al., 2013) | 363618 | 2162.10 hrs | No |
| MACS | 3537 | 9.85 hrs | No |
| SoundDescs (Koepke et al., 2022) | 10677 | 331.67 hrs | No |
| WavText5K (Deshmukh et al., 2022) | 2248 | 17.23 hrs | No |

### C.2 PRETRAINING SETTINGS

ONE-PEACE is a giant-size model with 4B parameters. We list the detailed hyper-parameters in Table 13. During pretraining, we introduce a lightweight Transformer decoder to recover the masked units from the visible units. The decoder is similar to the modality-fusion encoder, each block of it also consists of a shared self-attention layer and three modality FFNs. It has 2 layers with 768 hidden size, 2048 intermediate size, and 12 attention heads.

The pretraining of ONE-PEACE is divided into two stages: vision-language pretraining and audio-language pretraining. For vision-language pretraining, we pretrain ONE-PEACE for 200K steps with a batch size of 32768. We use the AdamW (Loshchilov & Hutter, 2019) optimizer with $(\beta_1, \beta_2) = (0.9, 0.98)$ and $\epsilon = 1e\text{-}8$. The peak learning rate is set to $5e - 4$, with a linear warmup of 3000 steps and a cosine decay scheduler. The image resolution is set to $256 \times 256$. The maximum text sequence length is set to 70. For regulation, we use weight decay with $0.05$ and disable dropout. We employ drop path (Huang et al., 2016) with a $0.4$ rate. For audio-language pretraining, we keep the model parameters related to vision and language (e.g., self-attention layers) frozen and only update the parameters that pertain to audio, such as A-Adapter and A-FFN. In this stage, we pretrain ONE-PEACE for 10 epochs with a batch size of 3072. The peak learning rate is set to $2e - 4$, with a linear warmup of 1 epoch and cosine decay scheduler. The maximum audio duration is set to 15s. For audio with a duration of less than 1s, we first repeat the input and then truncate it to 1s. Other hyper-parameters remain the same as vision-language pretraining.

## D DETAILS OF DOWNSTREAM TASKS

### D.1 VISION TASKS

Here we describe the implementation details of different vision tasks, including image classification (Russakovsky et al., 2015), semantic segmentation (Zhou et al., 2016), object detection (Lin et al., 2014), and video action recognition (Carreira & Zisserman, 2017). All detailed hyperparameters are listed in Table 14.

**Image Classification** We provide the fine-tuning results on ImageNet-1k (Russakovsky et al., 2015). Following recent studies in self-supervised learning for computer vision, we use global pooling of all image tokens excluding the class token, and append a LayerNorm with a linear layer for classification. To further unleash the potential of ONE-PEACE, we perform intermediate fine-tuning on ImageNet-21k (Deng et al., 2009). We set the label smoothing as 0.3 and do not use random erasing, mixup, and cutmix data augmentations. For fine-tuning on ImageNet-1k, we use exponential moving average (EMA) for model parameters and set the EMA decay rate as 0.9998. For intermediate fine-tuning on ImageNet-21k, we do not use EMA. We also use Zero Redundancy Optimizer (Rajbhandari et al., 2020) and set the stage as 1.

| #Layers | Hidden Size | Intermediate Size | Attention Size | #Parameters | | | | | | | |
|---|---|---|---|---|---|---|---|---|---|---|---|
| | | | | V-Adapter | A-Adapter | L-Adapter | V-FFN | A-FFN | L-FFN | Shared Attention | Total |
| 40 | 1536 | 6144 | 24 | 3.4M | 19M | 78M | 1.15B | 1.15B | 1.15B | 378M | 4B |

Table 13: **Detailed hyperparameters of ONE-PEACE model configuration.**

Table 14: **Fine-tuning setting for vision tasks.**

| Config | ImageNet-21k | ImageNet-1k | ADE20K | COCO | Kinetics 400 |
|---|---|---|---|---|---|
| Optimizer | | | AdamW | | |
| Optimizer momentum | | | $\beta_1, \beta_2$=0.9, 0.999 | | |
| Numerical precision | | | fp16 | | |
| Peak learning rate | 1e-4 | 5e-5 | 1.5e-5 | 1e-4 | 3e-4 |
| Layer-wise lr decay | 0.85 | 0.9 | 0.95 | 0.9 | - |
| Weight decay | 0.05 | 0.05 | 0.05 | 0.1 | 0.05 |
| Batch size | 5120 | 1024 | 16 | 64 | 64 |
| Warmup ratio | 0.375 | 0.2 | 0.0375 | 0.003 | 0.1 |
| Training epochs | 40 | 15 | 30 | 50 | 30 |
| Drop path | 0.4 | 0.4 | 0.5 | 0.6 | 0.4 |
| Image resolution | $256^2$ | $512^2$ | $896^2$ | $1280^2$ | $256^2$ |

**Semantic Segmentation**   We provide the fine-tuning results on ADE20k (Zhou et al., 2016). We use Mask2Former (Cheng et al., 2022) as the segmentation head. We first intermediate fine-tune segmentation head on coco-stuff (Caesar et al., 2018) dataset for 80k steps. The learning rate is set as 2e-5 and the rest hyperparameters are the same as ADE20K shown in Table 14. Then we fine-tune the model on ADE20K. Both experiments use the cosine learning rate decay scheduler.

**Object Detection**   We provide the fine-tuning results on COCO (Lin et al., 2014) with ViTDet (Li et al., 2022c). We use large-scale jitter (Ghiasi et al., 2021) data augmentation and fine-tune for 50 epochs. We use the linear learning rate decay scheduler and decay the learning rate at 44 and 48 epochs respectively.

**Video Action Recognition**   To perform video action recognition, following AIM (Yang et al., 2023), we freeze the parameters of the pre-trained model and add spatial and temporal MLP adapters in each transformer layer. We conduct experiments on Kinetics 400 (Carreira & Zisserman, 2017) dataset. Due to the invalid video links, there are many different versions of the K400 dataset and we use the version released on AcademicTorrents. We use the cosine learning decay scheduler and set the backbone learning rate multiplier of 0.1.

### D.2   AUDIO-RELATED TASKS

We describe the implementation details of audio-text retrieval, audio(-vision) classification, and audio question answering here. All detailed hyperparameters are listed in Table 15.

**Audio-Text Retrieval**   We evaluate ONE-PEACE on AudioCaps (Kim et al., 2019) and Clotho (Drossos et al., 2019) datasets. To get better results, we merge the training set of AudioCaps (Kim et al., 2019), Clotho (Drossos et al., 2019), and MACS as the fine-tuning dataset. We use the model to extract the features of audio clips and texts, and then calculate the cosine similarity between these features. The recall@k is employed as the evaluation metric.

**Audio Classification**   We conduct experiments on three datasets: ESC-50 (Piczak, 2015), FSD50K (Fonseca et al., 2020), and VGGSound (Chen et al., 2020a). ESC-50 is an environmental sound dataset that contains 2000 environmental audio recordings and 50 labels. We directly use the pretrained ONE-PEACE model to perform zero-shot audio classification on ESC-50. Specifically, we extract audio embeddings from the audio clips and text embeddings from the label names. Then we determine the labels of the audio clips by calculating the similarity between the embeddings. For

Table 15: **Fine-tuning setting for audio(-language) tasks.**

| Config | AudioCaps & Clotho | FSD50K | VGGSound | AQA / AVQA |
|---|---|---|---|---|
| Optimizer | AdamW | | | |
| Optimizer momentum | $\beta_1, \beta_2 = 0.9, 0.999$ | | | |
| Weight decay | 0.05 | | | |
| Gradient clip | 0.0 | | | |
| Warmup ratio | 0.1 | | | |
| Learning rate schedule | cosine decay | | | |
| Numerical precision | `bf16` | | | |
| Peak learning rate | 1.5e-4 | 1e-4 | 7e-5 | 7e-5 |
| Layer-wise lr decay | 0.95 | 0.9 | 0.95 | 0.9 |
| Batch size | 384 | 128 | 512 | 128 |
| Training epochs | 10 | 10 | 10 | 10 |
| Drop path | 0.9 | 0.5 | 0.6 | 0.5 |
| Max duration | 20s | 15s | 15s | 15s |
| Max frames | - | - | 16 | - / 12 |

the multi-label sound event dataset FSD50K, we input the original audio into the model and utilize multi-head attention pooling (MAP) to aggregate the features. We use BCELoss as the loss function and report the mean average precision on the test set. For the video-audio classification dataset VGGSound, we extract video clips of 8 seconds at a rate of 2 frames per second. These 16 frames are arranged in a $4 \times 4$ grid to form an image with a $480 \times 480$ resolution. We also extract audio clips from the videos, with a maximum duration of 15 seconds. Subsequently, we directly input the image and audio clips into the model and utilize MAP to aggregate the features, using cross-entropy as the loss function and reporting the accuracy on the test set.

**Audio(-Video) Question Answering** We conduct experiments on the AVQA dataset (Yang et al., 2022a). Each sample in this dataset consists of a video, a question, and four candidate answers. We extract video clips of 6 seconds at a rate of 2 frames per second. These 12 frames are arranged in a $3 \times 4$ grid to form an image with a $480 \times 480$ resolution. During training, we concatenate each answer with the audio, video, and question, and extract the features through the model. We then minimize the pairwise hinge loss between the positive features and negative features. To perform the audio question answering task, we extract audio clips from the videos and excluded the visual information.

### D.3 VISION-LANGUAGE TASKS

Here we describe the implementation details of different vision-language tasks, including image-text retrieval (Young et al., 2014; Lin et al., 2014), visual grounding (Yu et al., 2016; Mao et al., 2016), visual question answering (Antol et al., 2015), and visual reasoning (Suhr et al., 2018). All detailed hyperparameters are listed in Table 16.

**Image-Text Retrieval** We evaluate ONE-PEACE on MSCOCO (Lin et al., 2014) and Flickr30K (Young et al., 2014) datasets, and report the results on the widely used Karpathy test split (Karpathy & Fei-Fei, 2015). We use the model to extract the features of images and texts, and then calculate the cosine similarity between these features. The recall@k is employed as the evaluation metric.

**Visual Grounding** This task requires the model to locate an image region based on a text description. We conduct experiments on RefCOCO, RefCOCO+, and RefCOCOg datasets (Yu et al., 2016; Mao et al., 2016). The image and text are fed to the model simultaneously, then we use multi-head attention pooling (MAP) (Lee et al., 2018) to aggregate the features from all image patches. The pooled output is used to predict the continuous corner coordinates of the bounding box $(x_1, y_1, x_2, y_2)$, where $x_1$ and $y_1$ denotes the normalized top left coordinates, $x_2$ and $y_2$ denotes the normalized bottom right coordinates. We report the standard metric Acc@0.5 on the validation and test sets.

Table 16: **Fine-tuning setting for vision-language tasks.**

| Config | MSCOCO | Flickr30K | RefCOCO/+/g | VQA | NLVR2 |
|---|---|---|---|---|---|
| Optimizer | | | AdamW | | |
| Optimizer momentum | | | $\beta_1, \beta_2 = 0.9, 0.999$ | | |
| Weight decay | | | 0.05 | | |
| Gradient clip | | | 0.0 | | |
| Warmup ratio | | | 0.1 | | |
| Learning rate schedule | | | cosine decay | | |
| Numerical precision | | | bf16 | | |
| Peak learning rate | 8e-5 | 7e-5 | 1.5e-4 | 3e-4 | 1e-4 |
| Layer-wise lr decay | 0.9 | 0.9 | 0.9 | 0.85 | 0.9 |
| Batch size | 3072 | 3072 | 256 | 512 | 128 |
| Training epochs | 15 | 20 | 30 | 10 | 25 |
| Drop path | 0.5 | 0.4 | 0.4 | 0.5 | 0.4 |
| Image resolution | 432 | 432 | 384 | 768 | 256 |

**Visual Question Answering** This task requires the model to answer the question based on an image. We perform experiments on the VQAv2 dataset (Goyal et al., 2017). Following previous works (Wang et al., 2022b; Li et al., 2021; 2022a; Zeng et al., 2021), we use the training and validation set of VQAv2 for training, including additional question-answer pairs from Visual Genome (Krishna et al., 2017). The image and question are fed to the model simultaneously, then we use MAP to aggregate the features from all text tokens. The pooled output is fed into a classifier to predict the answer from the 3,129 most frequent answers. We report the final score on the test-dev and test-std sets.

**Visual Reasoning** Given a text and a pair of images, this task requires the model to distinguish whether the text truly describes the images. We conduct experiments on the NLVR2 dataset (Suhr et al., 2018). Following the common practice, We treat each sample as two image-text pairs, each containing a text and one image. Then we input these pairs into the model respectively. The final pooled outputs are concatenated together and fed to a classifier to predict the label. We report accuracy on the dev and test-P sets.

# E ADDITIONAL EXPERIMENTS

## E.1 EFFECTS OF PRETRAINED AUDIO FEATURE EXTRACTOR

We conduct a systematic analysis of the impact of the pretrained audio feature extractor. We find that although the parameters of the feature extractor are only 4.6M, accounting for only about 0.12% of the total parameters, it has a significant impact on the model performance. As shown in Table 17, the feature extractor with random initialization only achieves 85.8 accuracy on the ESC-50 dataset, while using pretrained feature extractors results in better performance. Notably, using the WavLM feature extractor can lead to the largest improvement (+6.0). We attribute this to the fact that WavLM is trained on a more diverse audio dataset compared to Hubert and Wav2Vec 2.0, making its feature extractor more suitable for environmental sound tasks.

Table 17: **Ablation studies of pretrained audio feature extractors.** We report zero-shot accuracy on the ESC-50 dataset.

| Feature Extractor | Random Init. | Hubert (Hsu et al., 2021) Init. | Wav2Vec 2.0 (Baevski et al., 2020) Init. | WavLM (Chen et al., 2021) Init. |
|---|---|---|---|---|
| ESC-50 Acc. | 85.8 | 89.6 (+3.8) | 90.0 (+4.2) | 91.8 (+6.0) |

## E.2 RESULTS ON VISION-LANGUAGE UNDERSTANDING TASKS

Table 18 presents the results of ONE-PEACE and baselines on two popular multimodal understanding tasks: visual question answering (VQA (Antol et al., 2015)) and visual reasoning (NLVR-2 (Suhr

et al., 2018)). For the VQA task, ONE-PEACE achieves a score of 82.6 on the test-dev set and 82.5 on the test-std set, outperforming previous strong baselines like CoCa and BLIP-2. For the NLVR2 task, ONE-PEACE surpasses CoCa with gains of 1.7 and 1.3 on the dev set and test-P set respectively. Notice that our results on both tasks are lower than BEiT-3. This may be attributed to two reasons: Firstly, BEiT-3 is pretrained on in-domain datasets such as MSCOCO (Lin et al., 2014) and Visual Genome (Krishna et al., 2017), which usually results in better downstream finetuning effects. Secondly, BEiT-3 incorporates pure text data for pretraining, which improves its language understanding ability and consequently enhances its multimodal understanding ability. In addition, OFA and BLIP-2 have shown that combined with language pretrained models can improve performance on multimodal understanding tasks. Therefore, we will explore the combination of ONE-PEACE and language pretrained models in the future.

Table 18: **Results on vision-language understanding tasks.** Without initialized with language pretrained models or pretraining on pure text data, ONE-PEACE outperforms the strong baselines Flamingo, CoCa and BLIP-2.

| Method | VQA | | NLVR-2 | |
| --- | --- | --- | --- | --- |
| | test-dev | test-std | dev | test-P |
| ALBEF (Li et al., 2021) | 75.8 | 76.0 | 82.55 | 83.14 |
| BLIP (Li et al., 2022a) | 78.25 | 78.32 | 82.15 | 82.24 |
| X-VLM (Zeng et al., 2021) | 78.22 | 78.37 | 84.41 | 84.76 |
| OFA (Wang et al., 2022b) | 82.0 | 82.0 | - | - |
| Flamingo (Alayrac et al., 2022) | 82.0 | 82.1 | - | - |
| CoCa (Yu et al., 2022) | 82.3 | 82.3 | 86.1 | 87.0 |
| BLIP-2 (Li et al., 2023b) | 82.2 | 82.3 | - | - |
| BEiT-3 (Wang et al., 2023) | **84.2** | **84.0** | **91.5** | **92.6** |
| ONE-PEACE | 82.6 | 82.5 | 87.8 | 88.3 |

### E.3 TRAINING CURVES OF DIFFERENT STRUCTURES

Figure 4 demonstrates the convergence performance of different architectures. Among all the architectures, the model with shared self-attention layers and separated FFNs exhibit the fastest convergence speed and the best convergence performance.

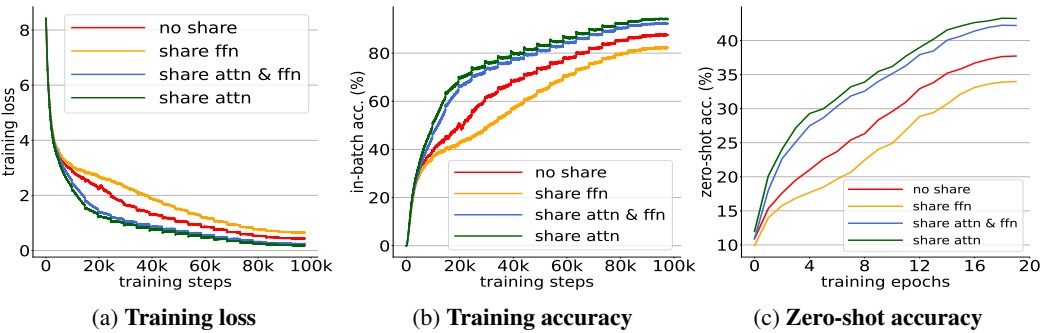

(a) **Training loss**  (b) **Training accuracy**  (c) **Zero-shot accuracy**

Figure 4: **Training curves of different structures.** The model with shared self-attention layers and separated FFNs ("share attn") outperforms other structures, exhibiting the fastest convergence speed and the best convergence performance. (The curve appears as a staircase shape because we use exponential moving average to calculate relevant indicators in each epoch.)

## F VISUAL GROUNDING EXAMPLES FOR OUT-OF-DOMAIN PICTURES

We test the visual grounding ability of ONE-PEACE on out-of-domain pictures. The model is fine-tuned on the RefCOCOg (Mao et al., 2016) dataset. We compared ONE-PEACE and OFA (Wang

et al., 2022b) on an Pokémon picture.[3] As shown in Figure 5, given a specific description of a Pokémon, both ONE-PEACE and OFA can obtain the correct result. However, when we directly provide the name of the Pokémon, OFA fails to obtain the correct result while ONE-PEACE can give a correct answer.

We further test ONE-PEACE with a more complex anime picture, *One Piece*. As shown in Figure 6, we ask ONE-PEACE to locate the characters based on their names. Although ONE-PEACE hasn't seen any anime pictures in the RefCOCOg dataset, it still achieves a recognition accuracy of 56.6%, which demonstrates its strong transferability.

## G   MORE EXAMPLES OF EMERGENT ZERO-SHOT RETRIEVAL

In this section, we provide more examples to demonstrate the emergent zero-shot abilities of ONE-PEACE, including audio-to-image, audio+image-to-image, and audio+text-to-image retrieval. The audios are selected from ESC-50 (Piczak, 2015), and the images are retrieved from ImageNet-1K (Deng et al., 2009) and MSCOCO (Lin et al., 2014). By reading this section, we hope that readers can better perceive ONE-PEACE.

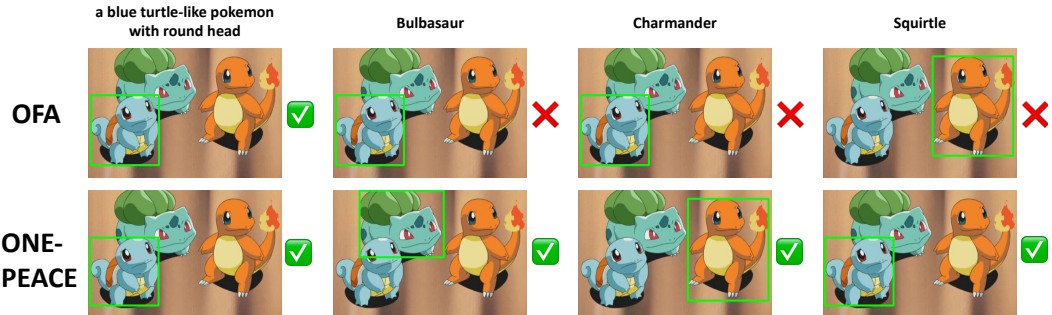

Figure 5: **Visualization of *Pokémon* in visual grounding task.** When given a specific description, both OFA and ONE-PEACE can give the correct result. However, OFA is unable to locate the correct region if we directly provide the name of the Pokémon, while ONE-PEACE can give a correct answer.

---

[3]We use the Hugging Face spaces demo of OFA: https://huggingface.co/spaces/OFA-Sys/OFA-Visual_Grounding

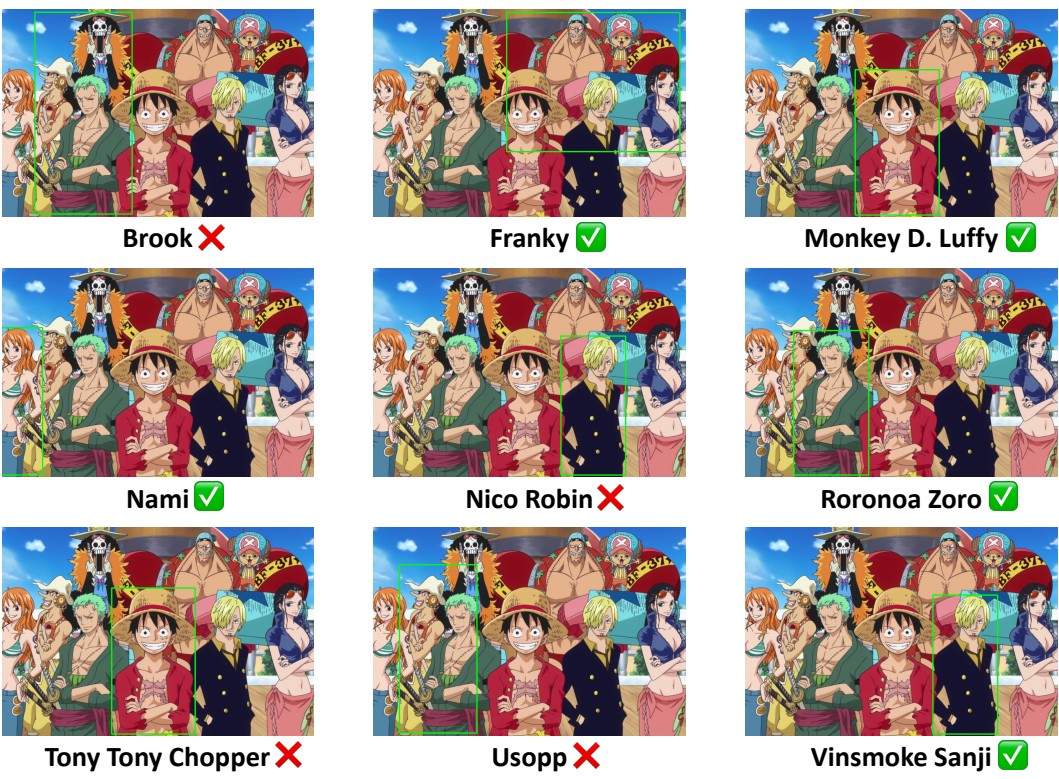

Figure 6: **Visualization of ONE-PEACE locating different characters of *One Piece*.** Given the names of 9 members of the Straw Hat Pirates, ONE-PEACE correctly located 5 of them from the picture.

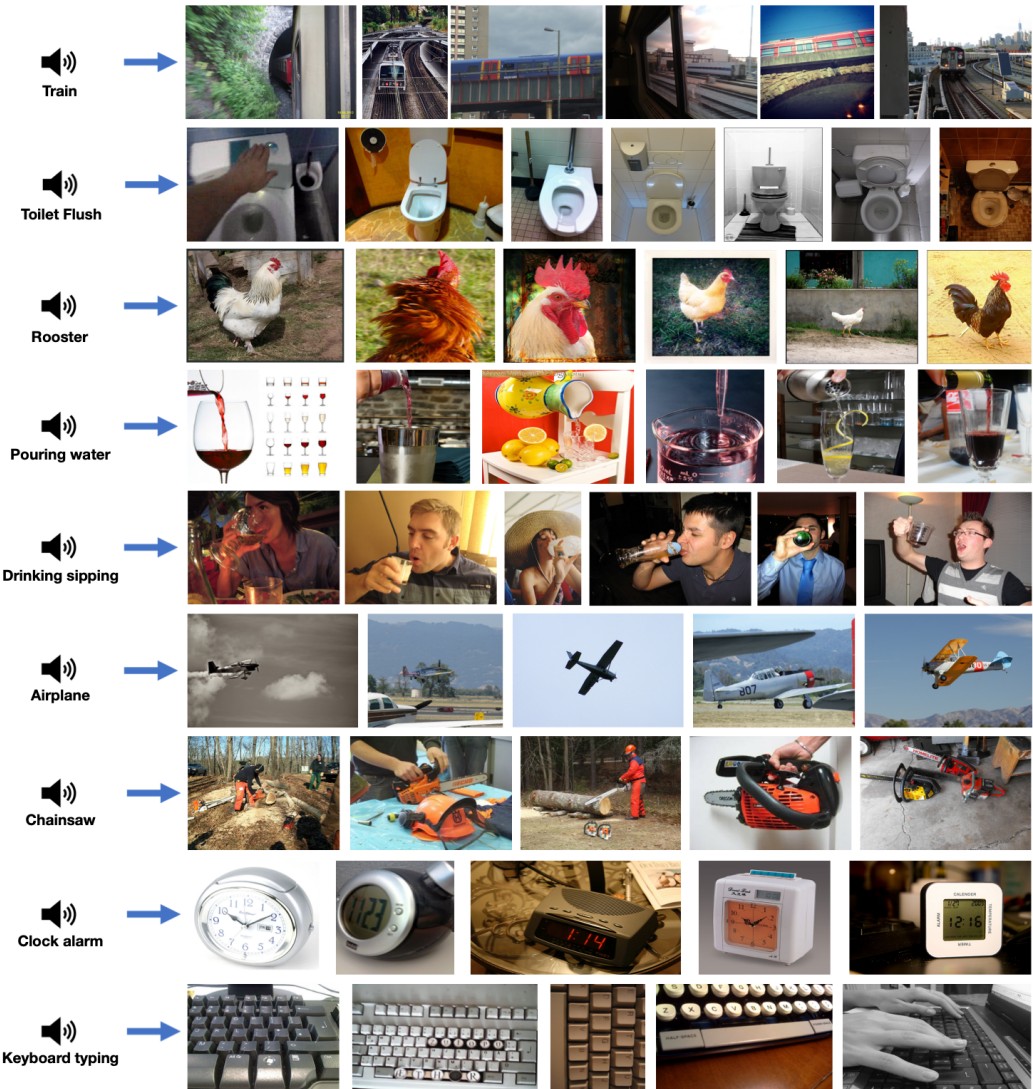

Figure 7: **Audio-to-image retrieval.**

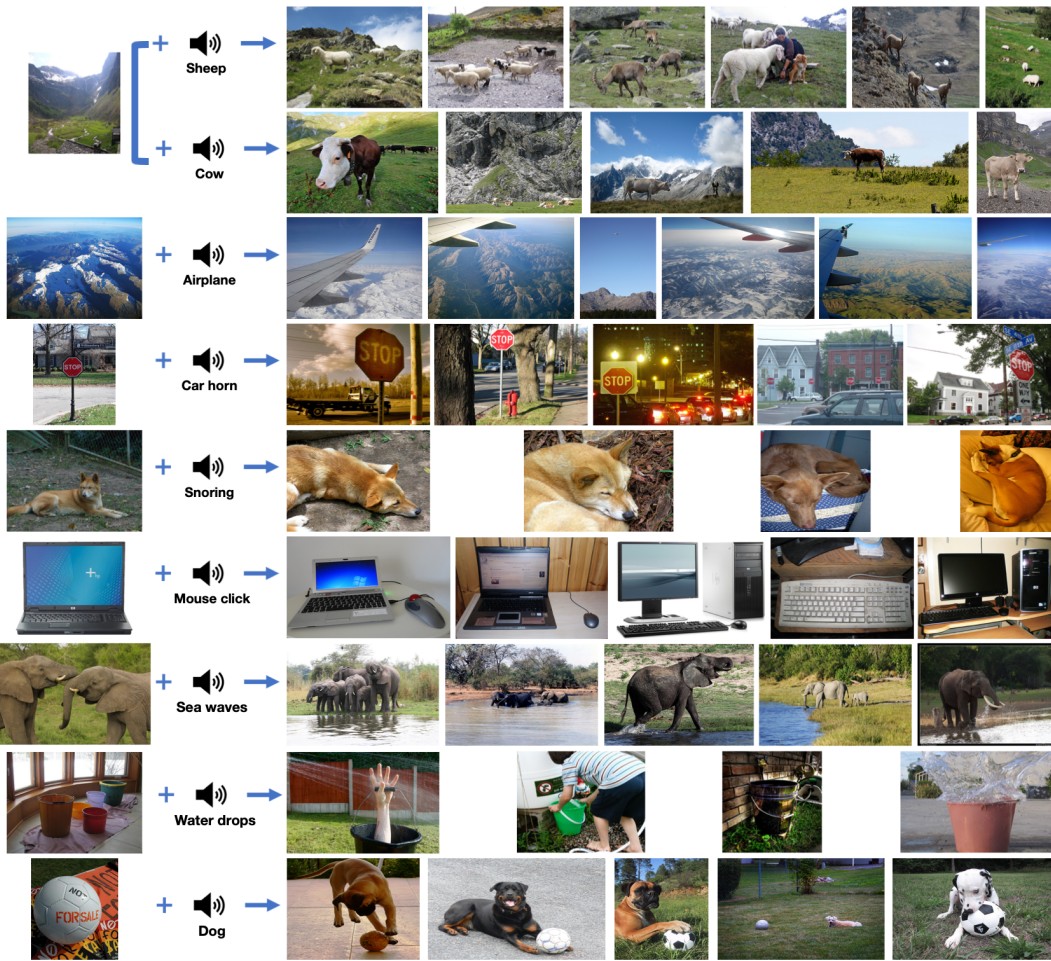

Figure 8: **Audio+image-to-image retrieval.**

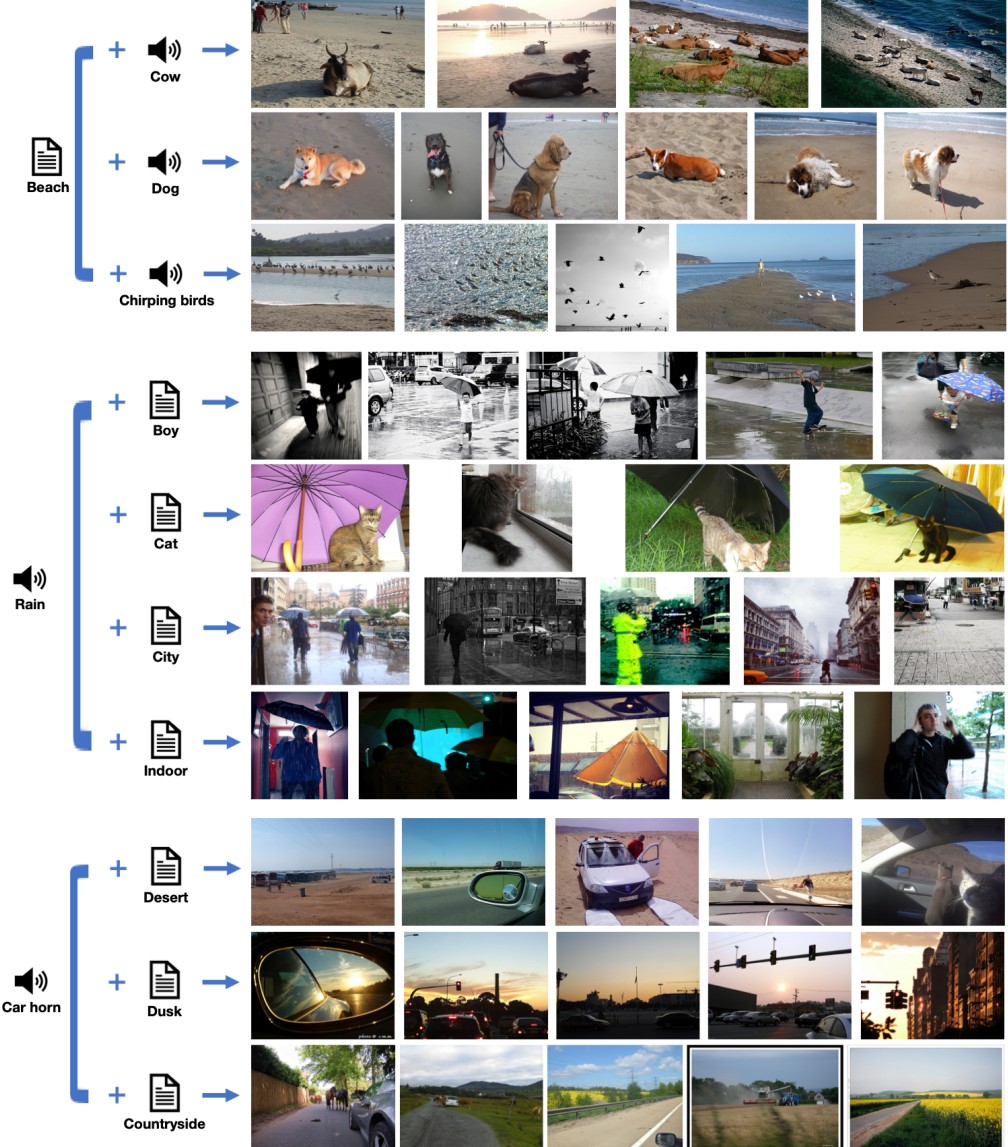

Figure 9: **Audio+text-to-image retrieval.**

