# OpenReview forum: "ONE-PEACE: Exploring One General Representation Model Toward Unlimited Modalities"
_ICLR.cc/2024/Conference — Submitted to ICLR 2024_

### Official Review · Reviewer_DHvY · 2023-10-30

**Soundness:** 4 excellent
**Presentation:** 3 good
**Contribution:** 2 fair
**Rating:** 6
**Confidence:** 4

**Summary:**

This work proposed a extensible multi-modal model named ONE-PEACE. The architecture of ONE-PEACE consists of multiple modality adapters which extract unified features from different raw signals, and a modality fusion encoder which facilitate information extraction between and within different modalities. To pretrain ONE-PEACE, this work used cross-modal contrastive learning and intra-modal denoising contrastive learning. The experimental results on different tasks across various modalities shows the advantages of the model.

**Strengths:**

+ The experiments in this work are very comprehensive, including extensive experiments on downstream tasks, ablation experiments, and visual results.
+ The experimental results in the paper unquestionably demonstrate the superior performance of the model. The excellent fine-tuning and zero-shot performance across various downstream tasks in the visual, language, and audio modalities makes this model an outstanding three-modal universal model.
+ The model has a relatively straightforward overall architecture. The functions of each module are easy to comprehend.

**Weaknesses:**

- As an engineering project, this work is exceptional, with the proposed model demonstrating superior performance and good reproducibility. However, as an academic research, this work does not bring interesting findings or questions. It appears more like a fusion of various well-established and effective techniques, like hMLP,  Sub-LayerNorm and LayerScale. The contribution of this work should be reconsidered.
- Experiments solely on vision, language and audio modalities cannot prove that the model can generalize to "unlimited" modalities. Many heterogeneous modalities are hard to collect paired data and align with a existing modality, such as sensors, tables or even proprioception [a]. An experiment on a more heterogeneous modality like IMU should be conducted at least.
- I also wonder why the AVQA dataset is merely used for AQA task? The model is trained on paired data of two modalities, thus the performance of the model on a task with all three modalities is important. This experiment should be conducted.


[a] P. P. Liang, Y. Lyu, X. Fan, J. Tsaw, Y. Liu, S. Mo, D. Yogatama, L.-P. Morency, and R. Salakhutdinov, “High-modality multimodal transformer: Quantifying modality & interaction heterogeneity for high-modality representation learning,” Transactions on Machine Learning Research, 2022.

**Questions:**

The authors should provide a better exposition of the contributions of this work, especially the problems that the model addresses, rather than solely emphasizing its superior performance. The above weaknesses should be concerned.

Figure. 1: Adaptor -> Adapter

---

> ### Author Response · Authors · 2023-11-20
> **Response to Reviewer DHvY (1/2)**
>
> Thanks for your time reviewing and providing the helpful suggestions for our paper. We address open questions and remarks below:
>
> > **Clarify our contributions**
>
> We appreciate your recognition of our work as an exceptional engineering project with superior performance and good reproducibility. In the last paragraph of the introduction section, we outline the contributions of our work, including: 1) a highly extensible model with scalable-friendly architecture and modality-agnostic tasks that have the potential to expand to unlimited modalities; 2) the first representation model to achieve new SOTAs across a wide range of uni-modal and cross-modal tasks; 3) strong emergent retrieval capabilities for aligning unpaired modalities, eliminating the need for paired data collection across different modalities.
>
> Here we elaborate on the first point to emphasize the technical contributions of ONE-PEACE:
>
> 1. Scalable-friendly architecture. ONE-PEACE processes different modalities through shared attention layers and separate Adapters/FFNs. When introducing a new modality, it only requires the addition of new adapters and FFNs. While previous works like VLMO [1] and BEIT-3 [2] have also explored this design, they primarily concentrate on enhancing model performance rather than highlighting the utilization for modality expansion.
>
> 2. Modality-agnostic tasks. We combine two modality-agnostic pretraining objectives: cross-modal contrast and intra-modal denoising contrast, enabling the model to achieve outstanding zero-shot retrieval performance and fine-tuning performance simultaneously. We also analyzed the impact of different denoising losses in Table 11. We found that denoising contrastive loss is more compatible with cross-modal contrastive loss than other losses, it can further improve the model's performance. These experiments have not been explored well in previous works.
>
> 3. With the help of scalable-friendly architecture and modality-agnostic tasks, we design a two-stage training method. It allows us to add new modalities (audio) into ONE-PEACE without affecting its performance on existing modalities (vision and language), and even enables the achievement of SOTA results in tasks related to the new modalities. This modality extension method is a significant contribution of our paper, and to the best of our knowledge, it has not been explored in previous works.
>
> > **Experiment on more heterogeneous modalities**
>
> The design of our architecture and tasks can effectively handle heterogeneous modalities with a lack of paired data.
>
> 1. Modality adapters are solely responsible for transforming the raw input into vectors, allowing us the flexibility to choose its architecture, such as CNN, RNN, and MLP, guided by previous works (e.g., HighMMT, ImageBind [3]). Therefore, this design is very friendly to heterogeneous modalities. In addition, the training curves in Figure 4 indicate that the model architecture of ONE-PEACE helps accelerate the convergence speed of the model and achieve better results.
>
> 2. Denoising contrastive loss is a modality-agnostic task that can be used on any modality, and it does not rely on paired data. The results in Table 10 show that combining denoising contrast and cross-modal contrast pre-training tasks can achieve better modality alignment effects, making this design suitable for modalities that lack paired data.
>
> We agree that experiments on more heterogeneous modalities can further demonstrate the potential of ONE-PEACE, but due to constraints during the response period, we have included a discussion in the Appendix. Additionally, we have also discussed related works, such as HighMMT, in this section.
>
> ```text
> [1] Bao H, Wang W, Dong L, et al. Vlmo: Unified vision-language pre-training with mixture-of-modality-experts[J]. Advances in Neural Information Processing Systems, 2022, 35: 32897-32912.
> [2] Wang W, Bao H, Dong L, et al. Image as a foreign language: Beit pretraining for all vision and vision-language tasks[J]. arXiv preprint arXiv:2208.10442, 2022.
> [3] Girdhar R, El-Nouby A, Liu Z, et al. Imagebind: One embedding space to bind them all[C]//Proceedings of the IEEE/CVF Conference on Computer Vision and Pattern Recognition. 2023: 15180-15190.
> ```

---

> ### Author Response · Authors · 2023-11-20
> **Response to Reviewer DHvY (2/2)**
>
> Thanks for pointing this out. We have re-collected the AVQA dataset, conducted experiments, and updated the results in Table 6. As shown in Table 6, ONE-PEACE also achieves excellent results in this task. Thus, we have validated the ability of ONE-PEACE in all combinations of modalities, including vision-language (visual grounding, image-text retrieval), vision-audio (video-audio classification), audio-language (audio-text retrieval, audio question answering), and vision-audio-language (AVQA).
>
> | Model              | Ensemble  |   AVQA   |
> |:-------------------|:---------:|:--------:|
> | HME [1]            | HAVF [8] |   85.0   |
> | PSAC [2]           | HAVF [8] |   87.4   |
> | LADNet [3]         | HAVF [8] |   84.1   |
> | ACRTransformer [4] | HAVF [8] |   87.8   |
> | HGA [5]            | HAVF [8] |   87.7   |
> | HCRN [6]           | HAVF [8] |   89.0   |
> | PSTP-Net [7]      |     -     |   90.2   |
> | **ONE-PEACE**      |     -     | **92.2** |
>
> ```text
> [1] Fan C, Zhang X, Zhang S, et al. Heterogeneous memory enhanced multimodal attention model for video question answering[C]//Proceedings of the IEEE/CVF conference on computer vision and pattern recognition. 2019: 1999-2007.
> [2] Li X, Song J, Gao L, et al. Beyond rnns: Positional self-attention with co-attention for video question answering[C]//Proceedings of the AAAI conference on artificial intelligence. 2019, 33(01): 8658-8665.
> [3] Li X, Gao L, Wang X, et al. Learnable aggregating net with diversity learning for video question answering[C]//Proceedings of the 27th ACM international conference on multimedia. 2019: 1166-1174.
> [4] Zhang J, Shao J, Cao R, et al. Action-centric relation transformer network for video question answering[J]. IEEE Transactions on Circuits and Systems for Video Technology, 2020, 32(1): 63-74.
> [5] Jiang P, Han Y. Reasoning with heterogeneous graph alignment for video question answering[C]//Proceedings of the AAAI Conference on Artificial Intelligence. 2020, 34(07): 11109-11116.
> [6] Le T M, Le V, Venkatesh S, et al. Hierarchical conditional relation networks for video question answering[C]//Proceedings of the IEEE/CVF conference on computer vision and pattern recognition. 2020: 9972-9981.
> [7] Li G, Hou W, Hu D. Progressive Spatio-temporal Perception for Audio-Visual Question Answering[C]//Proceedings of the 31st ACM International Conference on Multimedia. 2023: 7808-7816.
> [8] Yang P, Wang X, Duan X, et al. Avqa: A dataset for audio-visual question answering on videos[C]//Proceedings of the 30th ACM International Conference on Multimedia. 2022: 3480-3491.
> ```

---

> > ### Comment · Reviewer_DHvY · 2023-11-23
> >
> > Thanks for providing the response. I will keep my score, currently.

---

### Official Review · Reviewer_9Evr · 2023-11-01

**Soundness:** 3 good
**Presentation:** 3 good
**Contribution:** 3 good
**Rating:** 8
**Confidence:** 2

**Summary:**

The paper introduce ONE-PEACE, a simple but effective model for tri-modality representation learning. The proposed model use two stage training to align the visual acoustic and linguistic representation and it generalize well to downstream tasks. The paper is well written and the proposed method is reproduciable.

**Strengths:**

- the paper is intuitive, straight-forward and working as expected.
- The paper is well written and easy to follow. The paper provides enough details to reproduce the results.
- The results on downstream tasks are solid and convincing. Although not the SOTA as for now, but still strong enough.

**Weaknesses:**

1. The proposed method use the two stage training method, the idea behind it is to align the visual and acoustic information with linguistic representation. This is a practical way to pre-train the model but may lead to representation mis-alignment between visual and acoustic modalities. Consider to add more results to backup the visual-acoustic feature alignment quality.
2. The experimental results section is sufficient with different downstream results, but lacks the insights on the comparison against other LMMs, especially the ones with different designs.

**Questions:**

1. Please further discuss if the two stage training is a compromise of dataset and data quality, the training resources or it is designed intentionally.
2. I actually have hands on experience with ONE-PEACE. seems the visual-acoustic alignment is on and off, is this because of the dataset and data quality or the model design?

---

> ### Author Response · Authors · 2023-11-20
> **Response to Reviewer 9Evr (1/2)**
>
> Thanks for your time reviewing and providing helpful suggestions for our paper. First of all, we would like to clarify that, to the best of our knowledge, ONE-PEACE is still the SOTA in a lot of datasets, such as ADE20K, RefCOCO+, VGGSound, FSD50K, etc.
>
> We address open questions and remarks below:
>
> > **W1: Add more results to backup the visual-acoustic feature alignment quality**
>
> Some of the experiments done in the paper, as well as our newly added experiments, can address your concerns:
>
> 1. We conducted experiments on the VGGsound dataset, which requires the model to predict the category of samples based on the video and audio information. The results are listed in Table 6. It can be seen that ONE-PEACE achieved SOTA results on this task, even outperforming previous video pretraining models.
>
> | Model          | VGGSound |
> |:---------------|:--------:|
> | CAV-MAE [1]      |   65.9   |
> | MMT [2]          |   66.2   |
> | MAViL [3]      |   67.1   |
> | **ONE-PEACE**  |   68.3   |
>
> 2. Following the suggestion of Reviewer DHvY30, we conducted experiments on the AVQA dataset. This dataset consists of video, audio, and text modalities, and it requires the model to answer questions based on the video and audio information. We have updated the results in Table 6. ONE-PEACE still achieves leading results on the dataset.
>
> | Model         | AVQA  |
> | :------------ | :---: |
> | PSTP-Net [4]  | 90.2  |
> | **ONE-PEACE** | 92.2  |
>
> In addition to the above experiments, we also provide the multi-modal retrieval cases in Figure 3 and Figures 7-9 to demonstrate the visual-acoustic feature alignment quality of ONE-PEACE.
>
> > **W2: Lacks insights on the comparison against other LMMs**
>
> In fact, the baselines compared in the paper cover various model architectures (encoder-only, dual-encoder, encoder-decoder, etc.) and pre-training tasks (contrastive loss, masked prediction, etc.). However, since these works utilize different pre-training datasets and techniques, we focus our analysis on the ablation study to ensure a fair comparison between different architectures and pre-training tasks.
>
> 1. In Section 5.4, we conduct a comparative analysis of the ONE-PEACE architecture against alternative frameworks. We hypothesize that the shared attention layer facilitates the effective integration of multimodal data, while the separated FFNs enhance the model's ability to extract modality-specific information. This hypothesis is substantiated by the findings presented in Table 9. The training curves in Figure 4 also indicate that the design of the shared attention layer and separate FFN layer can lead to faster convergence speed and better convergence performance.
>
> 2. We further analyze the impact of various pre-training tasks. In comparison to exclusively utilizing cross-modal contrastive learning, the inclusion of denoising loss enables the model to learn fine-grained information within modalities, thereby leading to better fine-tuning and zero-shot retrieval performance. The results in Table 10 confirm this. Table 11 also provides a further comparison of different forms of denoising loss and validates that the denoising contrastive loss used in this paper is more compatible with cross-modal contrastive loss.
>
> Moreover, we also compared with OFA[5], a generative model based on the encoder-decoder structure in Figure 5. It can be seen that ONE-PEACE exhibits better visual grounding ability on out-of-domain pictures.
>
> ```text
> [1] Gong Y, Rouditchenko A, Liu A H, et al. Contrastive audio-visual masked autoencoder[C]//The Eleventh International Conference on Learning Representations. 2022.
> [2] Zhu W, Doshi K, Yi J, et al. Multiscale Multimodal Transformer for Multimodal Action Recognition[J]. 2022
> [3] Huang P Y, Sharma V, Xu H, et al. MAViL: Masked Audio-Video Learners[J]. arXiv preprint arXiv:2212.08071, 2022.
> [4] Yang P, Wang X, Duan X, et al. Avqa: A dataset for audio-visual question answering on videos[C]//Proceedings of the 30th ACM International Conference on Multimedia. 2022: 3480-3491.
> [5] Wang P, Yang A, Men R, et al. Ofa: Unifying architectures, tasks, and modalities through a simple sequence-to-sequence learning framework[C]//International Conference on Machine Learning. PMLR, 2022: 23318-23340.
> ```

---

> ### Author Response · Authors · 2023-11-20
> **Response to Reviewer 9Evr (2/2)**
>
> > **Q1: About the two-stage pre-training**
>
> We designed the two-stage pre-training intentionally to demonstrate that ONE-PEACE has the potential to expand to unlimited modalities. In the first pre-training stage (VL pre-training), vision and language representation are aligned from paired VL data. In the second pre-training stage (AL pre-training), we want the new modality audio to align with language and not influence the VL-aligned feature so we freeze the text-branch parameters. This approach proves that ONE-PEACE can introduce new modalities without influencing the performance of the existing modalities, and even achieve excellent performance in the new modality. If we directly conduct pre-training for the visual, textual, and audio modalities simultaneously, we would not be able to demonstrate the scalability of ONE-PEACE to new modalities.
>
> > **Q2: About using ONE-PEACE**
>
> We appreciate your experience with ONE-PEACE and the feedback you provided. The following points should be considered when using ONE-PEACE:
>
> * ONE-PEACE has a maximum audio duration limit of 15 seconds. Any audio longer than this will be cut off. Therefore, you should ensure that the input audio is not too lengthy and that its crucial information is within the first 15 seconds.
> * The audio training data for ONE-PEACE solely consists of environmental sound datasets (such as bird chirping, alarm clock ringing, etc.), the accuracy of the other types of audio may be somewhat compromised.
> * If you use our demo, please note that the demo only uses 60,000 images as the candidate set (ImageNet val set + MSCOCO val set), so there may be some audio clips without corresponding images in the candidate set.

---

### Official Review · Reviewer_zvUR · 2023-11-02

**Soundness:** 2 fair
**Presentation:** 2 fair
**Contribution:** 2 fair
**Rating:** 5
**Confidence:** 4

**Summary:**

This paper proposes a model with 4B parameters that aligns and integrates representations across vision, audio, and language modalities. Two pertaining tasks, cross-modal aligning contrast and intra-modal denoising contrast are developed to align the semantic space of different modalities.

**Strengths:**

the paper is well-written, and the experiments are thorough. The problem of unifying representations from multiple modalities is significant and the proposed approaches showed some potential in this direction.

**Weaknesses:**

The paper presents an ambitious effort to amalgamate multiple modalities into a singular embedding space, a concept previously explored in works such as ImageBindm (encompassing images, language, audio, depth, thermal, and IMU modalities), CLAP (audio and language), ULIP (3D, image, and language), and Chatbridge (audio, video, image, and language), but seems not thoroughly discussed and compared. Notably, this study posits the advantage of a scaling-friendly architecture, purportedly capable of incorporating an unlimited array of modalities. While this is a compelling proposition, the reviewer suggests that the paper could better substantiate this claim by integrating and examining a broader range of modalities. Such an expansion would more robustly demonstrate the architecture's potential and scalability, thereby providing a more comprehensive understanding of its capabilities in handling diverse and complex multimodal datasets.

**Questions:**

see the weakness

---

> ### Author Response · Authors · 2023-11-20
> **Response to Reviewer zvUR**
>
> Thanks for your time reviewing and providing helpful suggestions for our paper. We address open questions and remarks below:
>
> > **Discuss and compare with related works**
>
> Some of the works have been discussed in the related work section (ImageBind) or compared in the experiments section (CLAP). Due to the page limit, we added a new discussion section in the appendix to discuss the other works. Here, we briefly discuss the differences between ONE-PEACE and these works:
>
> * ImageBind aligns multiple separated models with a pre-trained visual model to integrate different modalities into a unified semantic space. In contrast, ONE-PEACE uses a single model to align multiple modalities and allows them to interact through the self-attention mechanism, thus being more effective in handling fine-grained multimodal tasks like visual grounding. We updated the zero-shot results of ImageBind on ESC-50 in Table 6, and it can be seen that ONE-PEACE achieves better performance on this dataset.
>
> | Model         | ESC-50 (zero-shot) |
> |:--------------|:------------------:|
> | ImageBind     |        66.9        |
> | **ONE-PEACE** |      **91.8**      |
>
> * CLAP uses contrastive learning to align audio and language. We compare ONE-PEACE and CLAP in Table 5 and Table 6, ONE-PEACE achieves better performance in all the audio-related tasks.
>
> * ULIP introduces the 3D modality through contrastive learning. However, since ONE-PEACE has not incorporated the 3D modality, we cannot directly compare it with ULIP. We discuss the related works in the newly added discussion section.
>
> * Chatbridge provides multimodal capabilities for LLM by incorporating representation models. Our paper mainly compares ONE-PEACE with representation models, where generative models that combine LLMs and representation models have not been included in the baselines yet. As a general representation model, ONE-PEACE can also be integrated into LLM to provide multimodal capabilities. We discuss this in the discussion section.
>
> > **Integrating and examining more modalities**
>
> it is noteworthy that ONE-PEACE has demonstrated remarkable performance on various video-related datasets, including K400, VGGsound, and AVQA. These results validates its capability across four modalities: image, audio, language, and video. We agree incorporating additional modalities such as 3D data and IMU can further demonstrate the potential of ONE-PEACE. However, due to time constraints during the response period, we added a discussion in the discussion section and left this for future research.

---

> ### Author Response · Authors · 2023-11-22
>
> Thank you again for taking the time to review our paper. We appreciate your feedback and hope that we were able to address your concerns in our response. As the deadline is nearing, please let us know if you have any further questions before the discussion period ends. We are glad to address your concerns.

---

> > ### Comment · Reviewer_zvUR · 2023-11-23
> >
> > Thank you for the response. I've read other reviews and rebuttals.

---

> ### Author Response · Authors · 2023-11-23
>
> Thank you very much for the reply. We would like to ensure that we have addressed your concerns. If you have no further questions, would you kindly consider raising the score?

---

> > ### Comment · Reviewer_DHvY · 2023-11-23
> >
> > Sorry, for this comment by authors. I think it's a little rude to directly ask for raising score.

---

### Author Response · Authors · 2023-11-22
**General Response**

We thank the reviewers for their constructive comments. As suggested by the reviewers, we have added the following additional experiments and discussion to the paper:

* Compared with ImageBind on ESC-50 (Rev zvUR)
* Added discussion on more related work (Rev zvUR)
* Added discussion on how to integrate ONE-PEACE and LLM (Rev zvUR)
* Added explanation about expanding more modalities (Rev DHvY and zvUR)
* Experiments on vision-audio-language task AVQA (Rev DHvY and 9Evr)
* Fixed typo Adaptor -> Adapter in Figure 1 (Rev DHvY)

More detailed answers are in the individual replies to the reviews.

---

### Meta-Review · Area_Chair_o5RY · 2023-12-05

**Metareview:**

The paper proposes a model to align and integrate representations from vision, audio and language modalities. The model architecture contains shared self-attention, modality adapters and FFNs, which has the potential to expand to unlimited modalities. Extensive experiments are performed on various uni-modal and cross-modal tasks. The proposed model achieves comparable or superior performance than state of the arts.

Though reviewers all acknowledge the comprehensive evaluations in the work, they have raised two major concerns: 1) the model architecture is the same as prior work such as VLMO which does not bring new findings or insights; 2) the paper highlights the method can generalize to unlimited modalities but only evaluates on three modalities. The rebuttal did not address these concerns well. Therefore, the AC recommends rejection.

**Justification For Why Not Higher Score:**

Explained in the metareview.

**Justification For Why Not Lower Score:**

N/A

---

### Decision · Program_Chairs · 2024-01-16

Reject